# Simultaneous Extraction of Four Antibiotic Compounds from Soil and Water Matrices

Alison M. Franklin [1,2,*], Danielle M. Andrews [1,3], Clinton F. Williams [4] and John E. Watson [1]

1   Department of Ecosystem Science & Management, Pennsylvania State University,
    University Park, PA 16802, USA; dma65@pitt.edu (D.M.A.); jackwatson@psu.edu (J.E.W.)
2   U.S. Environmental Protection Agency, Office of Research & Development, Cincinnati, OH 45268, USA
3   Department of Geology, University of Pittsburgh, 4107 O'Hara St., Pittsburgh, PA 15260, USA
4   US Arid Land Research Center, United States Department of Agriculture-Agricultural Research Service,
    21881 North Cardon Lane, Maricopa, AZ 85138, USA; clinton.williams@usda.gov
*   Correspondence: franklin.alison@epa.gov; Tel.: +1-814-883-0905

**Abstract:** The incidence of antibiotic resistance is on the rise and becoming a major health concern. Analyzing the presence of antibiotic compounds in the environment is critical for determining the potential health effects for humans, animals, and ecosystems. For this study, methods were developed to simultaneously isolate and quantify four antibiotics important in human medicine (sulfamethoxazole—SMX, trimethoprim—TMP, lincomycin—LIN, and ofloxacin—OFL) in water and soil matrices. For water analysis, different solid phase extraction (SPE) cartridges (Oasis HLB plus and Phenomenex Strata-X) were compared. The Oasis HLB Plus SPE cartridge provided the highest and most consistent recoveries with $118 \pm 5\%$, $86 \pm 4\%$, $83 \pm 5\%$, and $75 \pm 1\%$ for SMX, TMP, LIN, and OFL, respectively. For soil analysis, different pre-treatments (grinding and freeze-drying) and soil extraction methodologies (liquid-solid extraction and accelerated solvent extraction (ASE)) were compared. The ASE system resulted in the highest overall recoveries of SMX, TMP, LIN, and OFL with an optimal extracting solution of acetonitrile/water ($v/v$, 50:50, pH 2.8). When the soil was ground and freeze-dried, trimethoprim recovery increased and when soil was ground, but not freeze-dried, LIN and OFL recoveries increased, while sulfamethoxazole recoveries decreased when soil was ground and freeze-dried. Based on this research, matrix characteristics, especially pH, as well as the pKa's and functional groups of the antibiotics need to be carefully considered when attempting to extract antibiotic compounds from a water or soil environment.

**Keywords:** antibiotics; sulfamethoxazole; trimethoprim; lincomycin; ofloxacin; soil; water; extraction; solid phase extraction; liquid-solid extraction; accelerated solvent extraction; liquid chromatography; freeze-drying





## 1. Introduction

Antibiotic compounds are a large, diverse class of pharmaceuticals heavily relied upon to prevent, treat, and manage bacterial infections and diseases in both humans and animals. In an average year, approximately 10.5 million kilograms of antibiotics are sold and distributed for use in food-producing animals, and approximately 3.4 million kilograms are prescribed and sold for human treatment [1,2]. Given the large quantity of antibiotics utilized in animal husbandry and human healthcare, concern has arisen about antibiotics reaching the environment and possible negative ecological impacts, including, but not limited to, increased incidences of antibiotic resistance as well as alterations to microbial communities, which may impact important biogeochemical and degradation processes, and cycling of nutrients [3–7]. Therefore, studying the fate of antibiotics in the environment as well as determining environmental concentrations will allow assessment of potential impacts in ecological systems and human populations.

Antibiotics enter the environment by two main pathways: animal manure inputs onto land, including manure application, and release of wastewater treatment plant (WWTP) effluent or biosolids [8,9]. Typically, effluent is released into surface water bodies, such as streams or rivers. However, in arid regions, the reuse of treated effluent for irrigation purposes and groundwater recharge is becoming more common. Therefore, in many cases either from release of animal excrement or WWTP effluent/biosolids, antibiotics may be interacting with soil and, most likely, pass through the soil column as water percolates through the profile. Due to its innate biological, chemical, and physical properties and structure, soil can act as a natural filter, and antibiotics that pass through the soil profile may interact with soil particles and/or be degraded by soil microorganisms. Antibiotics may become adsorbed to the soil due to hydrostatic or hydrophobic interactions, hydrogen bonding, cation- exchange, bridging or even surface complexation, all of which would effectively remove the compounds from the soil water [10]. This natural filtering process in the soil profile may help prevent the antibiotics from, then, reaching the water table or reduce the potential for stream deposition via runoff.

To understand the interactions of antibiotics with the soil profile, extraction procedures are necessary to remove them from soil and water matrices. The extraction of antibiotics from soil samples is a difficult process. Each class of antibiotics possesses different properties that result in unique chemical and physical characteristics. In general, antibiotics are characterized by pKa's due to their structures containing a hydrophobic core and hydrophilic functional groups and are sensitive to alterations in pH. The behavior of antibiotics can change considerably within the environment due to shifts in pH that cause a change in protonation or ionization state. As a result, if extremely polar or nonpolar solvents are used for extraction, then incomplete extraction may result. To extract antibiotics efficiently, knowledge of their pKa's, pH of the sample, and hydrophobicity of the antibiotic is necessary.

A wide variety of methods have been employed to determine antibiotic compounds from soil and water samples. The methods utilized most frequently for soil extraction are liquid-solid extraction (LSE) [11,12], ultrasonic assisted extraction (UAE) [13–15], accelerated solvent extraction (ASE) [16–18], and pressurized liquid extraction (PLE) [19,20]. Sample clean up and concentration of water samples and soil extracts typically occurs via solid phase extraction (SPE) [18,21,22]. Liquid chromatography with tandem mass spectrometry (LC/MS-MS) is commonly used to determine the analytes of interest [11,13,14,18,20]. To date, the only standard methodology available for the simultaneous extraction of antibiotics is EPA Method 1694 [23]. However, EPA 1694 methodology has been developed for an entire suite of antibiotics and other pharmaceuticals and personal care products, so the recoveries of individual antibiotics are not necessarily optimal and may vary significantly based on the characteristics of the matrix (soil or water).

The aim of this research project was to develop optimal extraction protocols for the simultaneous extraction of four antibiotics commonly found in treated wastewater effluent, so that these antibiotics can be monitored in soil and water environments with greater confidence. For water analysis and soil extract sample cleanup/concentration, two different solid phase extraction (SPE) cartridges (Oasis HLB plus and Phenomenex Strata-X) commonly recommended for isolation of antibiotic compounds were compared. For the analysis of soil, two different pre-treatments of the soil (grinding and freeze-drying) and two extraction methodologies (LSE and ASE) were compared. Figure 1 outlines the general processing steps that the water and soil samples underwent throughout this study.

The four antibiotics selected for this study were sulfamethoxazole (SMX), trimethoprim (TMP), ofloxacin (OFL), and lincomycin (LIN) (Table 1). These antibiotics were chosen based on local and national prescription rates, public health risks, and potential ecological impacts. Sulfamethoxazole and TMP have significant importance, because in combination, they are an effective treatment for MRSA, methicillin-resistant *Staphylococcus aureus*. Ofloxacin is a representative fluoroquinolone antibiotic, a class of powerful antibiotics that are used in both human and veterinary medicine [24]. Lincomycin is an antibiotic used to treat

severe infections in people who cannot take penicillin drugs and is commonly used in veterinary medicine [25]. With antibiotic resistance on the rise and limited antibiotics to treat infections, influxes of key antibiotics into the environment may not only have negative ecological impacts but long-term ramifications for public health, especially if they reach water sources.

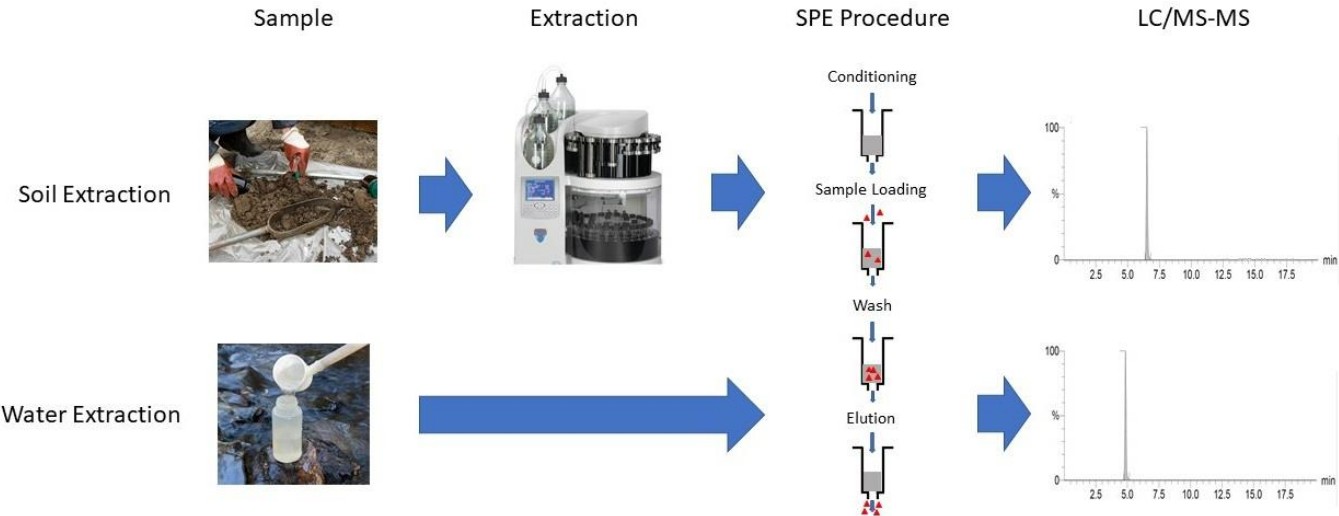

**Figure 1.** General analytical steps that water and soil samples underwent during this study to determine optimal extraction techniques to determine concentrations of four antibiotics (sulfamethoxazole, trimethoprim, ofloxacin, and lincomycin (SPE—Solid Phase Extraction, LC/MS-MS—Liquid Chromatography and tandem mass spectrometry).

**Table 1.** Characteristics of the four antibiotics (sulfamethoxazole, trimethoprim, ofloxacin, and lincomycin) extracted from soil and water.

| Antibiotics | Sulfamethoxazole | Trimethoprim | Ofloxacin | Lincomycin |
|---|---|---|---|---|
| Chemical Formula | $C_{10}H_{11}N_3O_3S$ | $C_{14}H_{18}N_4O_3$ | $C_{18}H_{20}FN_3O_4$ | $C_{18}H_{34}N_2O_6S$ |
| Chemical Structure | | | | |
| Molecular Weight (g mol$^{-1}$) [1] | 253.3 | 290.32 | 361.368 | 406.54 |
| pKa$_a$ | pKa$_1$: 1.4 pKa$_2$: 5.8 [2] | 6.6 [2] | pKa$_1$: 6.0 pKa$_2$: 8.0 [3] | 7.6 [2] |
| Log K$_{ow}$ [4] | 0.89 | 0.91 | −0.38 | 0.20 |

[1] [26], [2] [27], [3] [28], [4] [29].

## 2. Materials and Methods

### 2.1. Chemicals

Sulfamethoxazole (99.9%), TMP (99.7%), OFL (99.8%) and LIN-hydrochloride (≥95.0%) were used (Sigma-Aldrich, St. Louis, MO, US) for making standards and antibiotic solutions. Chlorpropamide (≥97.0%) (Sigma-Aldrich, St. Louis, MO, US) was used as an internal standard during quantification of antibiotic residues. Solvents used were methanol, MeOH (LC-MS grade), acetonitrile, ACN (LC-MS grade), 0.1% formic acid, FA, in water (LC-MS grade) and 0.1% FA in ACN (LC-MS grade) (Sigma-Aldrich, St. Louis, MO, US). Water used during this study was deionized (18 MΩ, pH 7). Sodium azide, NaN$_3$ (≥95.0%), (Sigma-Aldrich, St. Louis, MO, USA) and calcium chloride, CaCl$_2$ (>99%), (EMD Chemicals, Gibbstown, NJ, USA) were also used during method development.

### 2.2. Soil Characterization

Soil samples were collected for two purposes: (i) initial optimization of soil extraction procedures with a bulk soil and (ii) further evaluation of extraction procedures comparing four soils with different physical and chemical characteristics and different processing techniques. The bulk soil was an Ap horizon of a Hagerstown silt loam collected from the Penn State's Russell E. Larson Agricultural Research Farm at Rock Springs. This bulk soil was selected because it is ideal for agricultural use and can be found throughout the mid-northeastern United States [30]. This bulk soil had not received WWTP effluent irrigation nor manure application. The physical and chemical characteristics of this soil were determined at Penn State's Agriculture Analytical Services Lab (Table 2). The pH was determined by the water method with 1:1 soil:water ratio [31]; cation exchange capacity (CEC) was determined by summation [32]; and percent carbon (%C) was determined by combustion [33,34]. This bulk soil was utilized for the initial soil extraction optimization procedure that compared four different extracting solvents and two different extraction procedures.

**Table 2.** Properties of the soils used in the methodology development for the extraction of sulfamethoxazole, trimethoprim, lincomycin, and ofloxacin from soil.

| Soil Name | Location Land Use—Horizon | pH | Carbon (%) | Nitrogen (%) | EC (uS/cm) | CEC (cmol/kg) | Texture (%) | | | |
|---|---|---|---|---|---|---|---|---|---|---|
| | | | | | | | Sand | Silt | Clay | Classification |
| Bulk Soil | Rock Springs Cropped—Ap * | 5.8 | 1.14 | - | - | 14.2 | - | - | - | Silt Loam |
| Soil A | Rock Springs Cropped—Ap * | 6.5 | 0.89 | 0.097 | 79 | 25 | 22 | 59 | 18 | Silt Loam |
| Soil B | Rock Springs Cropped—B | 6.6 | 0.18 | 0.048 | 70 | 26 | 39 | 38 | 23 | Loam |
| Soil C | Astronomy Forested—A | 4.5 | 5.8 | 0.34 | 170 | 21 | 46 | 45 | 9 | Loam |
| Soil D | Astronomy Forested—B | 4.7 | 0.71 | 0.051 | 56 | 12 | 36 | 41 | 23 | Loam |

Key: EC—Electric Conductivity; CEC—Cation Exchange Capacity. * While Bulk Soil and Soil A were collected at Rock Springs cropped locations, these soils were collected at different time points and locations and, therefore, differed slightly in chemical and physical properties.

For experiments comparing extraction efficiencies between soil types and soil processing techniques, Hagerstown soil was collected from two locations and two depths to provide samples that varied in organic carbon (OC) content, CEC, pH, and texture. Soil samples from the A and B-horizons were collected from cropped and forested lands at Penn State's Rock Springs Research Farm and the Living Filter Astronomy site, respectively. These soils had not received effluent nor manure applications. These soils were labeled Soil A, an Ap horizon of cropped land classified as a silt loam; Soil B, the B horizon of cropped land classified as a loam; Soil C, the A horizon of a forested soil classified as a loam; and Soil D, the B horizon of a forest soil classified as a loam (Table 2). Soils A, B, C, and D were utilized for comparing extraction efficiencies based on (i) physical and chemical characteristics of the soil, (ii) treatment of soil prior to equilibration with antibiotic solution (air-dried, ground soils to pass through a 2.0 mm sieve versus field moist soils), and (iii) treatment of antibiotic-equilibrated soil prior to extraction (freeze-drying versus not freeze-drying soils).

For characterization of physical and chemical properties of Soils A, B, C, and D, subsets of each soil were air-dried, ground, and sieved to pass through a 2.0 mm sieve to determine pH, CEC, texture, %C, percent nitrogen (%N), and EC using standard techniques (Table 2). Texture was determined using the hydrometer method [35]. Electric conductivity and pH were measured using ratios of 1:2 and 1:1 of soil to water, respectively. Both parameters were measured using a Thermo Scientific Orion Star A215 pH/conductivity meter (Fisher

Scientific, Waltham, MA, USA). Cation-exchange capacity was measured using a standard unbuffered salt extraction method [36]. Subsamples of each air-dried, ground, and sieved Soils A, B, C, and D underwent additional grinding to pass through a 0.25 mm sieve. Triplicates of these subsamples were used for %C and %N analysis. Total OC and nitrogen were determined using a Carlo Erba CHNS-O Elemental Analyzer, EA1110 (Leco, MI, USA) with a solid sample module. The method used to prepare and analyze these soils was based on the Carlo Erba manual (CE Instruments, Wigan, UK, 1996), and due to previous work, no pretreatment steps were required to account for inorganic carbon in the soil samples.

### 2.3. Water Extraction

To determine simultaneous extractability of SMX, TMP, OFL and LIN from water samples, two commonly utilized solid-phase extraction (SPE) cartridges were compared to assess antibiotic recoveries from water samples that were either un-buffered (pH 7) or pH adjusted with 0.1% FA (pH 2.8). In preparation for the evaluation of each cartridge, deionized water (30 mL) with or without a pH adjustment was spiked with 127, 145, 181, and 203 $\mu$g L$^{-1}$ of SMX, TMP, OFL, and LIN, respectively (0.5 $\mu$M of each antibiotic compound). These spiked water samples were, then, passed through Oasis HLB (hydrophilic-lipophilic balance) Plus (Waters, Milford, MA, USA) and Phenomenex Strata-X polymeric (Torrance, CA, USA) reverse-phase SPE cartridges, both of which were recommended for extraction of antibiotics based on company literature. Both cartridges, regardless of pH adjustment, underwent the same SPE method modified from EPA 1694 methodology [22] and quantification method (*detailed under 2.5.1. Solid Phase Extraction (SPE) Procedure and 2.6. LC/MS/MS Quantification Procedure*).

### 2.4. Soil Extraction

### 2.4.1. Equilibration Procedure

The same equilibrium procedure was utilized for initial analysis of method development with bulk soil and subsequent analysis of soil characteristics and processing procedures using four different soils (Soils A, B, C, and D). Equilibration procedure consisted of soils (50 $\pm$ 0.02 g) being equilibrated with 100 mL of the antibiotic solution in a 200 mL glass centrifuge bottle to create a 1:2 ratio of soil to antibiotic solution. Equilibrations occurred for 24 h with shaking and rotation with centrifuge bottles covered in foil to prevent photodegradation. After equilibration, the bottles were centrifuged at 350 RCF for 1 h to separate the solid and liquid phases. The equilibrated antibiotic solution was decanted, and a subset (30 mL) was saved for later analysis (normally within 3 days) to determine the concentration of antibiotics adsorbed to the soil. Weights (mass balance) of soil alone, soil with the addition of antibiotic solution, and soil after removal of the antibiotic solution were used to account for the antibiotic solution still entrained within the soil matrix after equilibration, centrifugation and decanting. Spiked soils were then extracted using either the ASE or LSE method with a designated extracting solution within one hour following the equilibration solution being removed, except in the case of freeze-dried soils, which were analyzed within two months.

### 2.4.2. Extraction by LSE vs. ASE

For the simultaneous extraction of SMX, TMP, OFL, and LIN, four different extracting solutions and two different extraction methods were analyzed. These four antibiotics belong to the acidic group found in the EPA 1694 methodology [23] that outlines how to analyze certain pharmaceuticals and personal care products in environmental samples. Therefore, the soil extraction methodology for this study was designed with the intent of extracting these four antibiotics simultaneously, and each of the extracting solutions was acidified to a pH of 2.5–2.8.

The four extracting solutions were (i) 0.1% FA (pH 2.8), (ii) ACN/H$_2$O (*v/v*, 50:50, pH 2.8), (iii) MeOH (pH 2.5), (iv) MeOH/H$_2$O (*v/v*, 20:80, pH 2.5), which encompass various ratios of aqueous versus organic phases in order to find an extracting solution

that would work well for all of the antibiotics. Each of these extracting solutions was then used in conjunction with two extraction methods (ASE and LSE). The two extraction methods were analyzed to determine which system would be optimal and/or adequate for the extraction of the four antibiotics. Since ASE systems are not commonly available in most laboratory settings, finding an LSE method that would adequately extract antibiotics would provide a procedure that could be applicable universally. The extraction method and extracting solution that provided the highest overall recovery of each of the four antibiotics were, then, selected and utilized for subsequent soil extraction analysis using four different soil types and comparing a freeze-drying technique (*detailed under 2.4.3. Soil Characteristics and Processing Techniques Analysis*).

For each extraction method, triplicate samples of bulk soil were equilibrated with an antibiotic solution that consisted of deionized water containing 633 µg $L^{-1}$, 726 µg $L^{-1}$, 902 µg $L^{-1}$, and 1.01 mg $L^{-1}$ of SMX, TMP, OFL and LIN, respectively, (2.5 µM of each antibiotic) as well as 250 mg $L^{-1}$ (3.85 mM) of $NaN_3$ and 5.5 g $L^{-1}$ (0.05 M) of $CaCl_2$. Sodium azide was added to inhibit microbial activity, and $CaCl_2$ was added to counterbalance the addition of sodium with the $NaN_3$, which would otherwise affect the extractability of the antibiotics. These soils were then equilibrated following the procedure outlined in Section 2.4.1. Following equilibration, these antibiotic-spiked bulk soil samples (50 ± 0.02 g) had approximately 10 ± 5 mL of equilibration solution entrained within the soil particles.

Extraction by the LSE (shaker) method consisted of two rounds of shaking and rotating of saturated antibiotic-spiked bulk soil samples (in triplicate) with an extraction solution. Saturated antibiotic-spiked bulk soil (50 ± 0.02 g) and an extraction solution (100 mL) were added to the centrifuge bottles. Soils were equilibrated with the extraction solution for one hour with shaking and rotating, and the bottles were covered in foil to prevent photodegradation of the antibiotics. Bottles were centrifuged for one hour at 350 RCF to separate the liquid and solid phases. The extraction solutions were decanted with 98 ± 2 mL recovered. Subsets (30 mL) of the removed extraction solution were saved for later analysis. Soils were then extracted a second time by being equilibrated with 98 ± 2 mL of the extracting solution (to reach a 1:2 ratio of soil:extracting solution) for one hour with shaking and rotating, centrifuged, and decanted with a subset (30 mL) saved for later sample analysis. All saved extracted solutions were diluted so that the organic phase was <5% (*v/v*) and underwent an SPE process to clean up and concentrate the samples before they were analyzed by LC/MS/MS (*detailed under 2.6. LC/MS/MS Quantification Procedure*).

For the ASE procedure, approximately 15 ± 1.5 g of antibiotic-spiked soil was loaded into an extraction cell with 2 ± 0.02 g of sand on the top and the bottom of each cell. Extraction was performed using a Thermo Scientific Dionex ASE 350 Accelerated Solvent Extractor system (Fisher Scientific, Waltham, MA, USA). The cells were initially heated for 5 min to 100 °C. After heating, the cells underwent three 10 min static cycles, followed by a 60% volume flush, and a 100-s purge. Soils and cells were weighed at each step to account for mass balance. Volumes of the collected extracts ranged from 10 to 20 mL with an average of 14 ± 2 mL. Extract solutions were, then, diluted with 18 MΩ water so that the organic phase was <5% (*v/v*). Diluted extract solutions then underwent the SPE process to clean up and concentrate the antibiotic residues followed by quantification by LC/MS/MS (*detailed under 2.6. LC/MS/MS Quantification Procedure*). For the subsequent analysis of soil characteristics and processing techniques, Soils A, B, C, and D were extracted using the ASE system with ACN/$H_2O$ (*v/v*, 50:50, pH 2.8).

### 2.4.3. Soil Characteristics and Processing Techniques Analysis

Soils A, B, C, and D were spiked with SMX, TMP, LIN and OFL to determine the effect of different soil characteristics and soil processing techniques on extraction recovery. The four different soils (Soil A, Soil B, Soil C, and Soil D) were first air-dried, ground and sieved to pass through a 2.0 mm sieve. The soils were spiked with an antibiotic solution containing 6 µg $L^{-1}$ of SMX, 2 µg $L^{-1}$ of LIN and 4 µg $L^{-1}$ of TMP and OFL as well as 250 mg $L^{-1}$ (3.85 mM) $NaN_3$ and 5.5 g $L^{-1}$ (0.05 M) $CaCl_2$. Each soil (50 g of A, B, C, and

D) was equilibrating with 100 mL of spiked solution for 24 h. Once equilibrated solutions were removed, subsets of these samples were either extracted immediately or freeze-dried for approximately one week, then stored in the freezer (<0 °C) for one month to determine the effect of freeze-drying on antibiotic recovery.

### 2.5. Solid Phase Extraction

2.5.1. Solid Phase Extraction (SPE) Procedure

Solid phase extraction was used for the concentration and cleanup of water and soil extract samples as well as equilibrated antibiotic solutions before subsequent quantification of antibiotic residues. Large particulate matter was removed from the samples by vacuum filtration through 0.7 μM Whatman glass fiber filters (Pall Corporation, Ann Arbor, MI, USA). After filtration, samples underwent a SPE method modified from EPA 1694 methodology [22] to clean up and concentrate the residues using a reverse-phase SPE Oasis HLB Plus cartridge (Waters, Milford, MA, USA) and a vacuum manifold (Supelco, Bellefonte, PA, USA). Flow rates throughout the SPE procedure were held at 1–2 mL/min, unless otherwise specified. Conditioning the SPE cartridges included washing with MeOH (3 mL) (2x) and deionized water (3 mL) (2x), then discarding the eluents. Samples were loaded onto cartridges followed by one wash with deionized water (3 mL) with resulting eluents again discarded. Cartridges were dried under vacuum for approximately 5 min. Antibiotic residues were finally eluted with a 50:50 mixture of ACN and MeOH (3 mL) (2x), saving the eluent to be evaporated under a gentle stream of nitrogen ($N_2$) and heat (35 ± 5 °C). After evaporation, antibiotic residues were reconstituted in a 50:50 mixture of ACN and water (1 mL) spiked with 277 $\mu g \, L^{-1}$ (1 μM) of internal standard (chlorpropamide). The final reconstituted eluents were analyzed by LC/MS/MS.

2.5.2. Clean vs. Dirty Extraction Solutions with SPE

Analysis of clean versus dirty extraction solutions was conducted to account for how the soil matrix may alter the recovery of antibiotic extraction during the SPE process. Extraction of soils will result in extracts containing organic compounds, inorganic salts, colloidal clays, and possibly other materials that cannot be filtered out with a 0.7 um filter resulting in a more complex matrix of antibiotics than a solution of antibiotics made with clean water and/or other clean solutions. It is of interest to understand the extent to which such matrix effects may impact the recovery of antibiotics from the solution after soil extraction.

For this procedure, dirty extraction solutions were created by equilibrating each of the four extraction solutions for one hour with shaking with soil that had not been spiked with antibiotics using the same 1:2 ratio of soil (50 g) to extraction solution (100 mL) that was used for soil extraction method development. Triplicate samples of clean and dirty extraction solutions were analyzed after being spiked with 633 $\mu g \, L^{-1}$, 726 $\mu g \, L^{-1}$, 902 $\mu g \, L^{-1}$, and 1.01 $mg \, L^{-1}$ of SMX, TMP, OFL and LIN, respectively, (2.5 μM of each antibiotic). These spiked extraction solutions were thoroughly mixed and diluted with deionized water so that the organic content was less than 5% (*v/v*). These diluted extraction solutions were, then, run through the optimized SPE procedure to determine the percent recovery for the SPE portion of the soil extraction procedure. The final reconstituted eluents collected upon completion of the SPE process were analyzed by LC/MS/MS.

### 2.6. LC/MS/MS Quantification Procedure

Operating conditions for quantification were established and optimized by the Metabolomics Core Facility at Penn State. Analysis was performed on a Waters Xevo TQS, Ultra Performance (UP)LC®/MS/MS system. LC separation was achieved using an Acquity UPLC BEH C18 column (100 mm × 2.1 mm, 1.7 μm) (Waters, Milford, MA, USA) maintained at 40 °C. The mobile phase consisted of 0.1% formic acid in water as solvent A and 0.1% formic acid in acetonitrile as solvent B. Operating under a gradient method, the mobile phase initiated with a ratio of 97% of solvent A and 3% of solvent B. By 10 min,

the ratio was 55% of solvent A and 45% of solvent B. At 12 min, the ratio was 25% of solvent A and 75% of solvent B, which was held constant until 18 min, and then switched to initial operating conditions at 18.1 min. Total run time was 20 min. Flow rate was set at 250 μL/min with an injection volume of 1.0 μL for each sample.

Under these conditions, retention times for SMX, TMP, OFL, LIN, and chlorpropamide were 6.58 min, 4.64 min, 4.81 min, 4.38 min, and 10.21 min, respectively. The mass spectrometer was operating in positive electrospray (ESI+) mode with 24 multiple reaction monitoring (MRM) pairs, with Q3 working in SIM mode and each compound with a dwell time of 0.025 s. Three characteristic MRM transitions based on positive ion mass were selected for each antibiotic with the most intense characteristic transition utilized for quantification. Selected transitions for quantification were $254.07\ m/z > 156.01\ m/z$ for SMX, $192.13\ m/z > 123.07\ m/z$ for TMP, $362.17\ m/z > 261.08\ m/z$ for OFL, $407.17\ m/z > 126.13\ m/z$ for LIN, and $277.14\ m/z > 110.97\ m/z$ for chlorpropamide. Standard 5-point calibration curves were produced based on peak areas and known standard concentrations. Sample concentrations were estimated based on peak area outputs compared with calibration curve. Limit of quantification was determined on the machine for concentrated antibiotic residues in 1 mL of $ACN/H_2O$ were 2 μg $L^{-1}$, 1 μg $L^{-1}$, 1 μg $L^{-1}$, and 0.1 μg $L^{-1}$ for SMX, TMP, OFL, and LIN, respectively. The limit of detection based on the machine's capabilities was a signal to noise ratio of 3.

### 2.7. Statistical Analysis

One-way analysis of variance (ANOVA) and T-Tests were performed on recoveries obtained during SPE and soil extractions. Means values and standard deviations were calculated for all antibiotic concentrations recovered during methodology development. The software packages utilized for computation of statistics were SAS (version 9.2, SAS Institute, Cary, NC, USA) and Excel (version. 14.4.8, Microsoft for Mac 2011, Microsoft, Redmond, WA, USA).

## 3. Results

### 3.1. Water Extraction SPE Recoveries

Most recoveries of the four antibiotic compounds from aqueous samples (Table 3) fell within the quality control (QC) acceptance criteria for determination of pharmaceuticals and personal care products in water and soil matrices set by EPA Method 1694 [23], even the recoveries higher than 100%. The samples that were outside of these QC criteria had recoveries that fell below the acceptable range and included dirty MeOH and clean $ACN/H_2O$ extracts for TMP; clean MeOH, dirty MeOH, and clean $ACN/H_2O$ extracts for LIN; and HLB Plus (pH 7.0) and Strata X (pH 7.0) water samples and dirty MeOH, dirty FA, clean $ACN/H_2O$, dirty $ACN/H_2O$, and dirty $MeOH/H_2O$ extracts for OFL.

### 3.1.1. HLB plus vs. Strata-X SPE Cartridges

The HLB Plus without pH adjustment had the highest average recoveries for SMX, TMP, and LIN (118 ± 5%, 86 ± 4%, and 83 ± 5%, respectively), while HLB Plus with pH adjustment had the highest average recovery for OFL (75 ± 1.3%) (Table 3). These recoveries are comparable to or slightly higher than values found in the literature of 100%, 78%, 67%, and 82% for SMX, TMP, OFL, and LIN, respectively [17]. In addition to the HLB Plus having the highest overall recoveries, this cartridge provided more consistent results than the Strata X with coefficients of variation of 0.02–0.065 (2.0–6.5%) for the HLB Plus compared to 0.02–0.24 (2–24%) for the Strata-X.

**Table 3.** Average percent recoveries (±standard deviation) of four antibiotics extracted from water and clean versus dirty extracting * solutions using different pre-processing steps and cleanup procedures.

| Matrix | Pre-Process Step | Clean Up Procedures | Antibiotic Recoveries (%) | | | |
|---|---|---|---|---|---|---|
| | Matrix pH | SPE Cartridge | Sulfamethoxazole | Trimethoprim | Lincomycin | Ofloxacin |
| Water | 2.8 | HLB Plus | $101 \pm 13$ | $70 \pm 12$ | $65 \pm 3$ | $75 \pm 1$ |
| | 7.0 | HLB Plus | $118 \pm 5$ | $86 \pm 4$ | $83 \pm 5$ | $35 \pm 6$ |
| | 2.8 | Strata X | $106 \pm 5$ | $82 \pm 5$ | $58 \pm 2$ | $67 \pm 3$ |
| | 7.0 | Strata X | $103 \pm 20$ | $74 \pm 7$ | $62 \pm 4$ | $34 \pm 2$ |
| Clean MeOH * | 2.5 | HLB Plus | $92 \pm 7$ | $55 \pm 10$ | $2 \pm 0.2$ | $51 \pm 8$ |
| Dirty MeOH *¥ | | | $70 \pm 23$ | $46 \pm 8$ | $3 \pm 0.3$ | $47 \pm 4$ |
| Clean FA (0.1%) * | 2.8 | | $101 \pm 13$ | $70 \pm 12$ | $65 \pm 3$ | $71 \pm 1$ |
| Dirty FA (0.1%) *¥ | | | $104 \pm 10$ | $71 \pm 15$ | $83 \pm 4$ | $41 \pm 9$ |
| Clean ACN/$H_2O$ (50/50) * | 2.8 | | $70 \pm 15$ | $8 \pm 4$ | $0.3 \pm 0.1$ | $6 \pm 2$ |
| Dirty ACN/$H_2O$ (50/50) * ¥ | | | $97 \pm 6$ | $84 \pm 1$ | $31 \pm 9$ | $31 \pm 10$ |
| Clean MeOH/$H_2O$ (20/80) * | 2.5 | | $98 \pm 5$ | $77 \pm 8$ | $11 \pm 1$ | $62 \pm 5$ |
| Dirty MeOH/$H_2O$ (20/80) * ¥ | | | $101 \pm 3$ | $75 \pm 1$ | $23 \pm 2$ | $46 \pm 11$ |

Key: MeOH—Methanol, FA—Formic Acid, ACN—Acetonitrile * Extracting solutions were diluted with water to achieve a 5% volume prior to solid phase extraction. ¥ Term dirty extracting solution refers to an extracting solution that has interacted with a soil profile.

### 3.1.2. Clean vs. Dirty Extracts

When comparing different extracting solutions that were either clean or dirty (e.g., solutions that have or have not interacted with the soil profile, respectively) that underwent the SPE procedure, the percent recoveries for the antibiotics did vary by compound, type of extracting solution utilized, and, in some cases, whether the solution was clean versus dirty (Table 3). When 0.1% FA was the extracting solution and LIN was being determined, the dirty solution resulted in a significantly higher average recovery of $83 \pm 4\%$, while the clean solution had $65 \pm 3\%$ recovery. On the other hand, for OFL when using 0.1% FA, the clean solution resulted in an average percent recovery of $71 \pm 9\%$, which was significantly higher than the $41 \pm 7\%$ recovery with the dirty solution. For MeOH/$H_2O$ ($v/v$, 20:80) [pH 2.5] as the extracting solution, only LIN showed a statistically significant difference between dirty and clean samples with $24 \pm 1.7\%$ and $11 \pm 1.2\%$ average recoveries, respectively.

When ACN/$H_2O$ ($v/v$, 50:50) [pH 2.8] was used as the extracting solution, the dirty solution provided the highest recoveries for each of the antibiotic compounds (Table 3). Particularly with TMP, LIN, and OFL, the recoveries using dirty ACN/$H_2O$ as the extracting solution were statistically higher than the clean extracting solution with recoveries of $77 \pm 2\%$, $31 \pm 9\%$, and $31 \pm 10\%$, respectively, with the dirty solution and $8 \pm 4\%$, $0.31 \pm 0.1\%$, and $6.4 \pm 1.6\%$, respectively, with the clean solution. Although the recoveries for SMX were not statistically different, the dirty ACN/$H_2O$ had a higher average recovery with $91 \pm 5.6\%$ recovered versus $69 \pm 15\%$ recovered with the clean extracting solution.

### 3.2. Soil Extraction Results

### 3.2.1. Determination of Extraction Method

For the extraction of SMX, the optimal soil extraction procedure involved 50/50 ACN/$H_2O$ [pH 2.8] with the ASE system resulting in a percent recovery of $92 \pm 5.5\%$ (Figure 2, Table S1). For the other solvent mixtures using the ASE system, 0% was removed

with 0.1% FA, 79 ± 11% was removed with 20/80 MeOH/H$_2$O, and 81 ± 10% was removed with 100% MeOH. Whereas with the LSE procedure, 0% was removed using 0.1% FA in water, 34 ± 6% was removed using 20/80 MeOH/H$_2$O, 8 ± 1% was removed using 100% MeOH, and 73 ± 8% was removed using 50/50 ACN/H$_2$O.

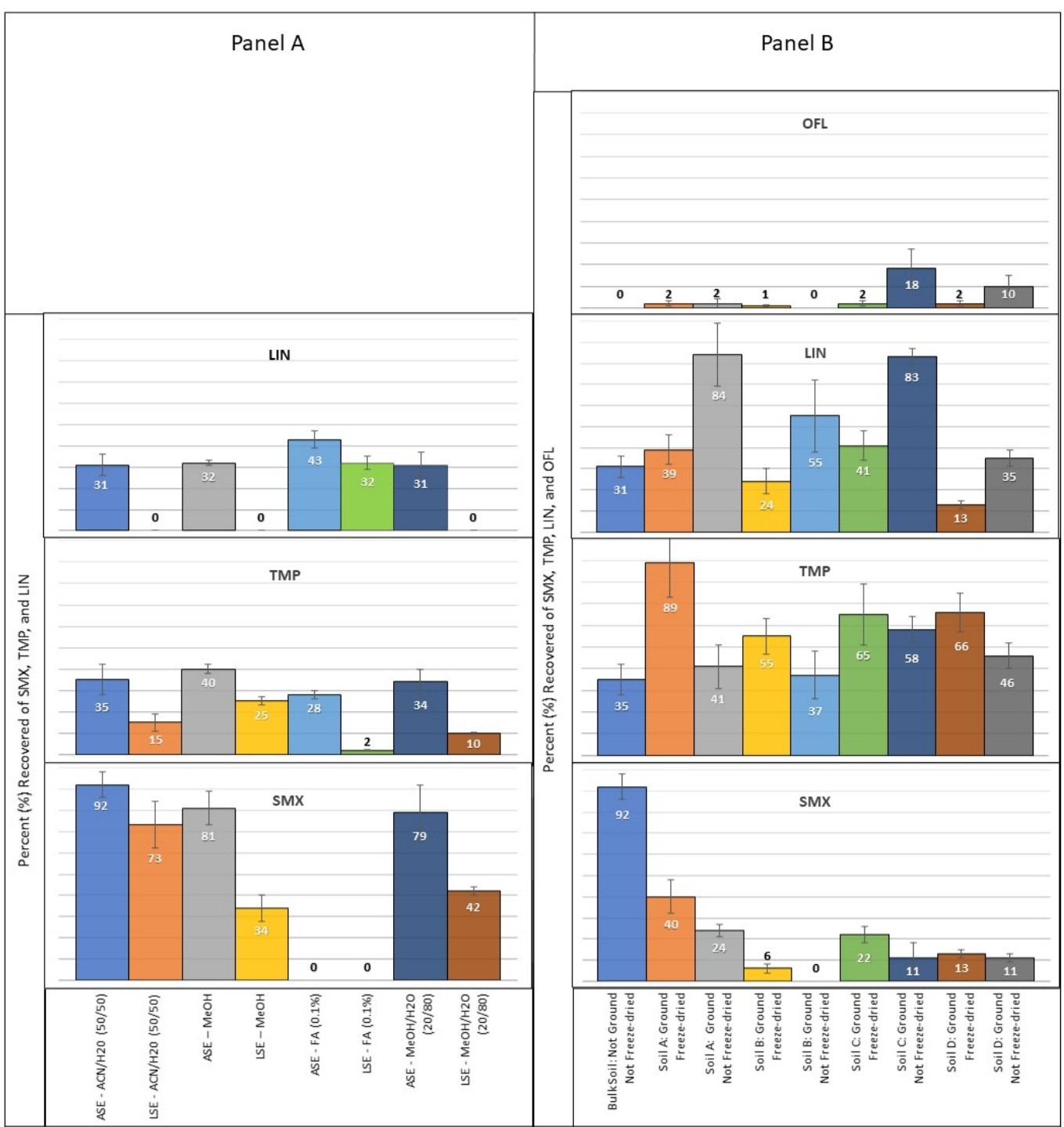

**Figure 2.** Average percent recoveries (±standard deviation) of sulfamethoxazole (SMX), trimethoprim (TMP), lincomycin (LIN), and ofloxacin (OFL) extracted from (**A**) bulk soil using accelerated solvent extraction (ASE) versus liquid solid extraction (LSE) with acetonitrile/water (ACN/H$_2$O) (50/50), methanol (MeOH), formic acid (FA) (0.1%), or MeOH/H$_2$O (20/80) and (**B**) bulk soil and four different soil types using ASE and ACN/H$_2$O (50/50) that underwent different pre-processing steps (grinding and freeze-drying).

For TMP extraction from bulk soil, using the ASE system, $28 \pm 2\%$ was removed using 0.1% FA in water, $34 \pm 6\%$ was removed using 100% MeOH, $40 \pm 1.9\%$ was removed using 20/80 MeOH/$H_2O$, and $35 \pm 7\%$ was removed using 50/50 ACN/$H_2O$. For LSE, $2 \pm 0.3\%$ was removed with 0.1% FA, $25 \pm 2\%$ was removed with 100% MeOH, $10 \pm 0.4\%$ was removed with 20/80 MeOH/$H_2O$, and $17 \pm 4\%$ was removed with 50/50 ACN/$H_2O$.

For extraction of LIN with the ASE system, $43 \pm 4.0\%$ was recovered using 0.1% FA in water, $32 \pm 2.2\%$ was recovered with 100% MeOH, $31 \pm 5\%$ was recovered with both 20/80 MeOH/$H_2O$ and 50/50 ACN/$H_2O$. Using LSE for extraction resulted in recoveries of $32 \pm 3.2\%$ with 0.1% FA in water, while 0% with 100% MeOH, 20/80 MeOH/$H_2O$, and 50/50 ACN/$H_2O$.

Ofloxacin was not recovered using any of the four extracting solutions or either of the extraction methods (Figure 2).

### 3.2.2. Soil Type and Processing Technique

Recoveries of SMX in the freeze-dried soils ranged from 14–44% with an average of $20 \pm 15\%$ and the highest quantity of SMX recovered from Soil A and the lowest amount recovered from Soil B. For non-freeze-dried soils, the recoveries of SMX ranged from 0–24% with an average recovery of $12 \pm 10\%$, and the highest amount recovered from Soil A and the lowest recovered from Soil B. The recoveries of TMP ranged from 55–89% with an average of $69 \pm 14\%$ with the highest quantity of TMP recovered from Soil A and the lowest quantity recovered from Soil B. Compared to the bulk soil analysis with a maximum recovery of $35 \pm 7\%$ with 50/50 ACN/$H_2O$, the recoveries with the ground and freeze-dried soils are significantly higher. In fact, even though the recoveries for the ground and not freeze-dried are lower than the freeze-dried soil, these recoveries ranging from 38–58% with an average of $46 \pm 9\%$ are either equivalent to or slightly better than the maximum recoveries of TMP from the bulk soil analysis (Figure 2, Table S1).

For LIN extractions, recoveries for freeze-dried soils had percent recoveries ranging from 25–48% with an average of $29 \pm 13\%$, which were similar to percent recoveries of the bulk soil analyzed previously ($31 \pm 5\%$). On the other hand, soils that were not freeze-dried had significantly higher recoveries that ranged from 35–84% with an average of $64 \pm 24\%$ recovered, which was overall higher than the recoveries obtained from the bulk soil.

The recoveries of OFL with freeze-drying were extremely low ranging from 1–2% recovered with an average of $2 \pm 0.5\%$. However, with non-freeze-dried soils, in soils C and D, recoveries of OFL were $18 \pm 9\%$ and $10 \pm 5\%$, respectively, while in soils A and B, the recoveries were similar to the freeze-dried soils with $2 \pm 2\%$ and 0% extracted, respectively. Without grinding the soil, recovery of OFL was consistently 0%.

### 3.2.3. Matrix Effects Based on Internal Standard Signal

Overall, the internal standard (chlorpropamide) signal was consistent across most of the soil samples. However, for some soil samples, the internal standard signal did demonstrate that signal suppression or enhancement was evident with significantly lower or higher signals, respectively. A clear trend was not apparent, with signal suppression and enhancement occurring with different soil types and typically with only one replicate of a particular soil sample, but not with the other replicates of that same sample.

## 4. Discussion

### 4.1. Water Extractions

#### 4.1.1. Comparison of SPE Cartridges

Based on the water extraction results (Table 3), the HLB Plus was determined to be the optimal cartridge for cleaning up the samples and isolating SMX, TMP, LIN and OFL due to the higher overall recoveries of the compounds with lower variability compared to the StrataX cartridge. While SMX, TMP, and LIN did not require a pH adjustment, OFL did require a pH adjustment for recoveries that were similar to the other antibiotic compounds. Therefore, analysis of the four compounds together is not ideal. When possible, OFL should

be analyzed separately. Ofloxacin, most likely, requires a pH adjustment due to having two pKa values of 6 and 8 meaning that at the water pH of 7, OFL would be zwitterionic. Adjusting to a pH around 2.8 would ensure that OFL was a positively ionized species and not possess both anionic and cationic functional groups. Since antibiotics that are grouped in the same class would have similar chemical properties and structures, it is reasonable to assume that the HLB Plus would be the most appropriate SPE cartridge for other antibiotic compounds in the same antibiotic classes as SMX, TMP, LIN and OFL (i.e., sulfonamides, lincosamide, and fluoroquinolones). In addition, other sulfonamide and lincosamide antibiotic compounds would probably not require a pH adjustment, but fluoroquinolones would require a pH adjustment below the lowest pKa of that compound so that the compound of interest is in the anionic phase instead of zwitterionic.

4.1.2. Comparison of Clean vs. Dirty Soil Extraction Solutions

Analysis of clean extracting solutions spiked with antibiotics compared to extracting solutions that are spiked with antibiotics after interactions with soil, termed dirty, was performed to comprehend the potential impact of the extracting solution (i.e., not a pure aqueous solution) and the soil matrix on the SPE recoveries of antibiotic compounds during a soil extraction procedure. The range of recoveries for each compound varied greatly when comparing the recoveries from different extracting solutions, regardless of whether the solution was clean or dirty. Of note, the highest recoveries for each antibiotic compound with a specific extracting solution were similar to the highest recoveries obtained during the SPE process with pure aqueous solutions.

In most instances, whether the extracting solution was dirty or clean had little impact on the recoveries of the antibiotic compounds as significant differences in the percent recoveries of these four antibiotic compounds were not noted between most of the clean versus dirty solutions. However, in some instances, the dirty solutions had higher recoveries than the clean solutions. For ACN/ $H_2O$ (*v/v*, 50:50) [pH 2.8], interaction of that solution with the soil (resulting in a dirty extraction solution) had a significant effect on the recoveries with the dirty solutions resulting in higher recoveries for all the antibiotics compared to the clean extracting solution (i.e., extracting solutions that had never interacted with soil) (Table 3). Notably, the recoveries of TMP, OFL, and LIN from the clean ACN/ $H_2O$ solution were significantly lower than the recoveries from the dirty ACN/ $H_2O$ solution, ranging from 0.3–8% and 31–84%, respectively. The minimal recoveries from the clean ACN/ $H_2O$ solution for these three antibiotics could be due to these compounds preferring the aqueous phase instead of the solid matrix in the SPE cartridge. Acetonitrile is a strong organic solvent commonly employed for eluting chemical compounds from SPE cartridges [37,38]. Even though the ACN/ $H_2O$ (*v/v*, 50:50) solution was diluted to a 5% organic phase, the presence of ACN in the solution could still impact the interactions of the antibiotics with the solid matrix of the SPE cartridge. Furthermore, the recovery of LIN was higher in all the dirty extraction solutions compared to the clean solutions, demonstrating that under the conditions of the clean extracting solutions, this compound preferred the aqueous phase compared to the SPE cartridge. However, in the dirty solutions, the presence of soil particulate matter could have played a role with the antibiotics preferentially interacting with the soil particles. The soil particles would have been captured in the solid phase of the SPE cartridge resulting in retention of the antibiotic compounds. Matrix suppression or enhancement was not a likely cause of these differences since the recoveries of the internal standard were consistent across the samples.

Understanding the influence of the extraction solution and the soil matrix on extracting solutions is an important consideration for determining and interpreting soil extraction recoveries for an antibiotic compound of interest. Extracting solutions need to be selected carefully not only based on the chemical properties of the antibiotic compound, but also, in certain instances, considering the interactions of the extracting solution with soil material itself and the possible impacts on antibiotic recoveries.

*4.2. Soil Extractions*

4.2.1. Determination of Extraction Method

Based on the percent recoveries for each of the extraction methods and extracting solutions for the four antibiotics, the ASE was selected as the optimal system and 50/50 ACN/H$_2$O (*v/v*) [pH 2.8] was selected as the extracting solution. This extracting solution would provide both the organic phase for removal of SMX and TMP with average recoveries of 92 $\pm$ 5.5% and 35 $\pm$ 7%, respectively, while also having an aqueous phase for the removal of LIN with an average recovery of 31 $\pm$ 5%. For each of the extracting solutions, the ASE had the highest recoveries for each antibiotic compared to LSE, except for OFL. In almost all cases, the recoveries with the ASE were significantly higher than the recoveries with the LSE system.

For the selection of the extracting solution, a single extracting solution did not offer the highest recoveries for all four of the antibiotics with the ASE (Figure 2, Table S1). In the case of SMX, 50/50 ACN/H$_2$O [pH 2.8] did provide the best recovery. The recovery of TMP with 50/50 ACN/H$_2$O, although not the highest recovery for the compound, was relatively close to the optimal recovery with 100% MeOH. Finally, the recovery of LIN with 50/50 ACN/H$_2$O is not optimal; however, the best extracting solution for this compound was 0.1% FA in water, and this extracting solution would not remove SMX. Therefore, even though 50/50 ACN/H$_2$O is not the best extracting solution for all four of the antibiotics, it is the most appropriate compromise for extracting these three antibiotics simultaneously. Ofloxacin was not considered during this decision since extraction of this compound was not possible with any of the systems or extracting solutions.

Sulfamethoxazole

After bulk soil was spiked with the equilibration solution containing 2.5 $\mu$M of SMX, approximately 30% of SMX was adsorbed to the soil. Based on SMX's second pKa (5.7) and the bulk soil pH (5.8), the percentage of SMX in the neutral state would be approximately 33–50%, which correlates very closely with the amount of SMX that was soil adsorbed, most likely due to hydrophobic interactions. The rest of SMX would be in the anionic state and repelled by the negative soil particles. Based on how closely soil adsorption relates to the percent of SMX in the neutral state, soils with a pH lower than the second pKa would have more SMX adsorbed to soil particles, while soils with a pH higher than SMX's second pKa would result in less of the compound interacting with the soil profile.

Extracting solutions with a higher ratio of organic to aqueous appear to be necessary for the extraction of SMX, especially since SMX at a low pH would be predominately neutral. In its neutral state, SMX would be driven by hydrophobic interactions and prefer organic and solid phases to aqueous phases. Acetonitrile at a maximum ratio of 50% in aqueous solution had the highest extraction capability even compared to 100% MeOH. Most likely, the reason that an extracting solution with ACN at a lower organic ratio than 100% MeOH had a higher extracting strength is that ACN has a higher capability of interrupting hydrogen bonds, which may be one of the major mechanisms of SMX soil adsorption. Methanol, on the other hand, due to its hydroxyl group may participate in hydrogen bonding rather than disrupt them, which would lower extraction capability. These results correlate with past literature for soil extraction of SMX, where recoveries using an ASE system or sonification with ACN as the extracting solution resulted in 92–96% recoveries of the compound [39,40].

Trimethoprim

To extract TMP, the best soil extraction procedure included 100% MeOH and the ASE system, which resulted in 40 $\pm$ 7% recovery (Figure 2, Table S1). Once the bulk soil had been equilibrated for 24 h with the extracting solution containing approximately 2.5 $\mu$M of TMP, on average, 94% of the compound was absorbed to the soil. Based on the pKa range for TMP (6.6–7.2) and the soil pH of 5.8, TMP would predominately be in the neutral state (90–100%) with only a small percentage in the cationic state; therefore, the compound

would preferentially interact with the organic and solid phases in the soil environment. Soils with a low pH would result in TMP predominately being adsorbed to the soil particles due to hydrophobic interactions.

The ASE system provided the best overall recoveries for TMP from soil with each of the extracting solutions when compared to LSE. Although 100% MeOH with the ASE system provided the highest overall recovery, this percent recovery was not significantly higher than the other three extracting solutions when using the ASE system, especially the extracting solutions with a higher organic phase. When considering LSE, however, only 100% MeOH produced appreciable recovery of TMP. Therefore, for the extraction of TMP, extracting solutions with higher organic phases are necessary to remove this predominately neutral compound from the soil matrix, especially when a high pressure and temperature system, such as the ASE, is not available.

Although the necessity of an extracting solution with a high organic phase correlates with past literature for the soil extraction of TMP (Figure 2, Table S1), the overall recoveries of TMP during this study were significantly lower than past extraction studies [41,42] where extractions with sonification and ultrasonification systems resulted in 63–77% recoveries. Based on the higher recoveries with sonification, TMP may be forming complexes with soil particles, establishing strong interactions with sorption sites, or becoming sequestered in pore spaces and, therefore, requiring physical disruption to remove the antibiotic compound from the soil matrix [43]. As a result, for optimal extraction of TMP, sonification may be necessary to disrupt any interactions or physical barriers that would prevent TMP removal. However, if a sonification system is not available, ASE and LSE systems will provide recoveries similar to this study, if not higher, with the use of an extracting solution with a high organic fraction.

Lincomycin

The method with the highest recovery for the extraction of LIN was an extraction solution of 0.1% FA in water with the ASE system, which resulted in an average recovery of 43 ± 4.0% (Figure 2, Table S1). After the soil was equilibrated for 24 h with equilibration solution containing 2.27 µM of LIN, approximately 80% was adsorbed to the soil. With its pKa of 7.6, LIN would predominately be in its cationic state (97%) in the bulk soil that had a pH of 5.8. LIN as a cation would be attracted to the negative soil particles, which explains why approximately 80±% of LIN was adsorbed to the soil. Similar to SMX and TMP, the ASE system provided the best extraction recoveries of LIN from the soil matrix regardless of the extracting solution utilized. However, even though recoveries were lower with LSE, this method provided insight concerning the optimal type of extracting solution to remove this compound from the soil matrix. Since only 0.1% formic acid provided removal of LIN during LSE, a pH adjusted aqueous solution is necessary to adequately remove this compound from the soil matrix, which is then confirmed by 0.1% FA in water providing the highest recovery of LIN with the ASE system. This assessment that pH is a driving force of LIN interactions with the soil matrix as well as its preference for the aqueous phases has been confirmed by previous researchers developing extraction protocols and studying sorption and desorption behaviors of this compound [44,45]. Since LIN would predominately be a cation in most soils, an aqueous phase would be adequate for removal of this compound from the soil matrix.

Ofloxacin

For analysis of bulk soil extractions, OFL was not recovered using any of the four extracting solutions or either of the extraction methods (Table S1). Previously, Christian et al. [17] was able to extract 30–50% of OFL from sand and sandy loam soils using an ASE system with 50/50 MeOH/$H_2O$. Nevertheless, given the high sand content with minimal number of sorption sites and low CEC content, extraction of OFL from sandy soils would be easier than extraction from the silt loam bulk soil used in this study that would have a higher number of potential sorption sites and CEC. Vazquez-Roig et al. [46] analyzed OFL extrac-

tion with a clay loam, which would have a higher sorption capacity similar to a silt loam, and extracted 28% of OFL; however, this study also used pressurized liquid extraction (PLE) instead of ASE with $Na_2$-EDTA as a chelator to interrupt soil-antibiotic interactions. Based on previous research, the chemical structure with a fluorine atom and behavior of the compound leading to strong interactions and complexes with aluminum and iron oxides of soil particles [47,48], a different extracting solution would be necessary to remove OFL from the soil. Most likely the addition of a chelator (EDTA) would be necessary to interrupt and prevent the strong interactions that occur between the chemical compound and the soil particles, and ammonia might need to be added to outcompete and remove OFL from the sorption sites in the soil. These soil extraction results again demonstrate that OFL would require a separate isolation and extraction procedure for optimal recovery and cannot be analyzed simultaneously with SMX, TMP and LIN.

### 4.2.2. Soil Type and Processing Technique
Sulfamethoxazole

For the extraction of SMX from Soils A-D that were equilibrated with an antibiotic solution containing 6 µg $L^{-1}$ of the compound, the highest recoveries for each soil type were obtained when the soils were freeze dried before extraction (Figure 2, Table S1). Given that SMX requires a higher organic phase for optimal extraction of the neutral compound from soil, the higher recoveries with the freeze-dried soils can be explained by the fact that soil moisture would not be present in the freeze-dried soils and, therefore, would not add additional aqueous phase. Whereas the soils that were not freeze-dried would contain water entrained within the soil matrix due to the equilibration procedure. This additional aqueous solution between the soil pores would effectively dilute the organic phase of the extracting solution during the ASE procedure. Since the ASE cell only contains a certain volume, if water is held within the soil particles that means less extracting solution and organic phase can be added to the cell to effectively extract the compound.

The recoveries for SMX with any of the ground and sieved soils regardless of freeze-drying are significantly lower than the average recovery ($92 \pm 5.5$%) obtained during initial methodology development using 50/50 ACN/$H_2O$ to extract SMX from bulk soil. Even though Soil A shares characteristics with the bulk soil, grinding and sieving the soil appears to negatively impact the ability to extract SMX after equilibration. Most likely, this lowered recovery is due to exposing additional surface area on the soil particles and allowing SMX to interact with additional soil particle surfaces that would otherwise be inaccessible in unground soil [27,42].

Even though grinding and sieving the soil lowered the percent recovery compared to the unground bulk soil, freeze-drying of ground and sieved soil did increase the recovery of SMX compared to similarly treated soil that was not freeze-dried. Therefore, one treatment that could increase overall percent recovery of SMX would be to leave the soil unground (no grinding and sieving), then freeze-dry it prior to extraction so that water would be removed from the soil and not dilute the extracting solution during the ASE procedure.

Trimethoprim

With the extraction of TMP from four different soils (Soils A-D), recoveries based on equilibration with antibiotic solution containing 4 µg $L^{-1}$ of TMP showed that freeze-dried soils had the highest overall recoveries (Figure 2, Table S1). Similar to SMX for the ground and sieved soils with comparisons with regard to freeze-drying, TMP followed a similar trend of freeze-drying increasing the overall recovery of the compound. Given that optimal TMP recovery relies on an organic phase present in the extracting solution to pull the neutral compound from the soil particles, with freeze-drying the extracting solution would not be diluted by water entrained within the soil particles. Whereas with soil that is not freeze-dried, the entrained water would lower the overall percent of the organic phase and lower overall recoveries [49].

However, unlike SMX, with the soil being ground and sieved, TMP recoveries increased compared to the bulk soil instead of decreasing. This behavior infers that interaction with the soil surface area is not the main driving force for TMP adsorbing to or remaining in the soil profile. As was observed in previous research, TMP may become sequestered in the pore space [42]. Grinding the soil appears to have the same effect as sonicating the soil before extraction, since the recoveries with grinding and freeze-drying the soil are comparable to and even better than the recoveries found in the literature that ranged from 65–77%. Consequently, for peak recoveries of TMP if sonification is not possible, grinding the soil in combination with freeze-drying would result in optimal recoveries of the compound.

Lincomycin

The general trend for higher recoveries of LIN during extractions with non-freeze-dried soils further confirms that LIN in the cationic state preferentially goes into the aqueous phase (Figure 2, Table S1). With water still entrained in the soil profile when freeze-drying is not employed, this excess water effectively increases the aqueous phase of the extracting solution and increases the recovery of LIN. Since LIN is predominately cationic within the different soils, with the increased surface area and exchange sites within the four different soils, LIN is expected to interact more strongly with the soil particles. However, the fact that grinding the soil increased the extractability of LIN when additional aqueous phase is present implies that LIN may have interactions with soil aggregates. The most likely scenario is the molecule becoming entrained in the soil pores of larger aggregates when soil is not ground and sieved prior to extraction.

When only comparing recoveries between the four different types of soils and neglecting comparisons of freeze-drying versus not freeze-drying, soils A and C have similar recoveries that are significantly higher than soils B and D. Therefore, based on the characteristics of the soils analyzed, percent organic carbon is most likely not a driving force in the interactions of LIN with the soil profile, because soils A and C have significantly different percent carbon, 0.89 and 5.8, respectively. However, when comparing between soils A and C versus soils B and D, the amount of clay differs in that soils A and C have a lower amount of clay particles compared to soils B and D. Even though the textural analyses are similar between all four soils, small changes in clay can have significant effects within a soil profile due to the high surface area of the clay particles and ability to attract cationic compounds. At the pH of 2.5 for the extracting solution, LIN with a pKa of 7.6 would be cationic; therefore, it would prefer the aqueous phase but in the presence of increased clay content the interactions with the negative clay particles with the antibiotic may be stronger than the compound's preference for the aqueous phase of the extracting solution.

Therefore, for optimal extraction of LIN, grinding the soil and not freeze-drying it would most likely result in the highest recoveries. By breaking up the soil aggregates, LIN is easier to extract from the soil matrix. Finally, for the optimal recovery of LIN, a pH adjustment below its pKa and aqueous extracting solutions are necessary to significantly remove the compound from the soil matrix.

Ofloxacin

While OFL was actually able to be recovered from Soils A-D, the recoveries were still very low compared to SMX, TMP, and LIN (<20%) (Figure 2, Table S1). However, the slightly higher recoveries that occurred with non-freeze-dried soils C and D could be pH dependent. With a pH of approximately 4.5 in each of those soils combined with the extracting solution with a pH of 2.8, OFL with a $pKa_1$ of 6.0 would be predominately cationic. As a result, the compound may have some preference for the aqueous phase, especially if at that low pH the compound is competing with hydrogen ions for sorption. Not freeze-drying the soil before extraction results in increased water entrained within the soil matrix that would effectively increase the aqueous phase of the extracting solution. However, even though the extracting solution is pH adjusted, that pH adjustment may

not significantly lower the overall pH of a soil that has a higher pH, such as Soils A and B with pH values of approximately 6.5. These soils (A and B) did not see an increase in the recovery of OFL when the soils were ground but not freeze-dried. Therefore, considering the pH of the soil may be critical for successful extraction, and additional pH adjustments may be necessary to fully extract the compound.

Finally, grinding the soil may or may not have an impact, because for soil A, which is similar to the bulk soil utilized during initial methodology development, the recoveries in either situation for freeze-dried versus not freeze-dried for soil A (2%) were similar to the bulk soil (0%). The marginal increase from 0% to 2% is not significant ($p > 0.05$). The main factor that seemed to impact the recovery was the pH of the soil and ensuring that a higher aqueous phase was then present in an extracting solution with low pH to remove the cationic OFL species. However, even when recoveries of OFL increased, those recoveries ranging from 10–18% were still lower than recoveries in the literature (Figure 2, Table S1), where 25–50% of the compound was recovered. Therefore, interactions with the negative soil particles are still strong forces that need to be overcome for the extraction of OFL, and as stated previously, a chelator and/or the incorporation of ammonia would most likely be necessary for optimal recovery of this compound.

### 4.2.3. Matrix Effects

Matrix effects can alter the recovery of antibiotic compounds by limiting recovery during the SPE process and/or causing a lower response (ion suppression) or enhanced response (ion enhancement) during LC/MS-MS analyses [50]. Utilization of the SPE process to clean up samples following soil extraction helps to minimize matrix effects; however, these effects may still occur, especially with samples with high levels of natural organic matter like soils. Natural organic matter can complex with the antibiotic compounds and/or compete for sorption sites on the SPE column preventing antibiotic-sorbent interactions. The matrix, essentially, reduces the number of free sites available for the retention of the antibiotic compounds on the SPE column and interferes with the antibiotic recovery. Furthermore, the matrix components that are coeluted with the antibiotic compounds can lead to an alteration of ionization efficiency of the target analytes, which are observed as ion suppression or ion enhancement [51].

For this study, the possible impacts of matrix effects were assessed by addition of an internal standard (chlorpropamide) prior to analysis via LC/MS-MS. While matrix effects of ion suppression and ion enhancement were observed in some samples with low and enhanced responses of chlorpropamide's signal, respectively, these observations were not consistent. For example, matrix effects were observed in one of the triplicate samples for one soil sample, but not for the other two. Additionally, matrix effects were not observed consistently for all samples of the same soil type. Overall, for most of the soil samples, chlorpropamide's signal was consistent and matrix effects did not play a major role in the determination of the antibiotic compounds.

### 5. Conclusions

Overall, each of the antibiotics was successfully extracted from the water and soil matrixes, except for OFL from soil matrixes where extraction recoveries were limited. For both water and soil extractions, one method was not sufficient to optimally extract SMX, TMP, LIN, and OFL simultaneously. Although SMX, TMP, and LIN could be extracted simultaneously with adequate recoveries, OFL would require a different extraction method for both water and soil matrixes to recovery appreciable quantities of the antibiotic. The HLB Plus was the optimal cartridge for water extraction procedures compared to the Strata-X, and while SMX, TMP, and LIN would not require a pH adjustment OFL would. For soil extractions, the ASE system with 50/50 ACN/$H_2O$ provided the best overall extraction recoveries for SMX, TMP, and LIN simultaneously; however, for significant recoveries of OFL, a different extraction method, possibly including a chelator and/or the incorporation of ammonia would be necessary. Higher recoveries occurred with TMP,

LIN, and OFL when the soil was dried, ground and sieved prior to equilibrating with the antibiotic-spiked solution. For SMX and TMP, if the soil was freeze-dried after equilibration with the antibiotic solution prior to extraction using the ASE system, average extraction recoveries were higher in freeze-dried soils compared to non-freeze dried. In the case of LIN, non-freeze-dried soils resulted in significantly higher average recoveries compared to freeze-dried soils. Therefore, even though SMX, TMP, LIN, and OFL are listed in EPA methodology 1694 under the acidic group, in order to obtain optimal extraction recoveries for each of these antibiotics, different water and soil extraction protocols would be necessary for each antibiotic. Furthermore, the extractability of each of these compounds is strongly correlated with the pH of the soil or water system as well as the type of extraction solution utilized.

**Supplementary Materials:** The following supporting information can be downloaded at: https://www.mdpi.com/article/10.3390/separations9080200/s1, Table S1: Average percent recoveries (±standard deviation) of four antibiotics extracted from different soil matrices using different pre-processing techniques, extraction solutions, and extraction methods.

**Author Contributions:** Conceptualization, A.M.F., D.M.A., J.E.W. and C.F.W.; methodology, A.M.F., D.M.A. and C.F.W.; software, A.M.F.; validation, A.M.F., D.M.A., J.E.W. and C.F.W.; formal analysis, A.M.F.; investigation, A.M.F. and D.M.A.; resources, J.E.W. and C.F.W.; data curation, A.M.F.; writing—original draft preparation, A.M.F.; writing—review and editing, A.M.F., C.F.W., D.M.A. and J.E.W.; visualization, A.M.F. and C.F.W.; supervision, D.M.A. and J.E.W.; project administration, J.E.W.; funding acquisition, J.E.W. All authors have read and agreed to the published version of the manuscript.

**Funding:** This study was partially funded by The Pennsylvania State University's Office of Physical Plant. J.E. Watson is supported, in part, by the USDA National Institute of Food and Agriculture Federal Appropriations under Project PEN04574 and Accession number 1004448. The present work was partially developed within the framework of the Panta Rhei Research Initiative of the International Association of Hydrological Sciences (IAHS).

**Institutional Review Board Statement:** Not applicable.

**Informed Consent Statement:** Not applicable.

**Data Availability Statement:** The data supporting the findings of this study are available within the article and supplementary Table S1.

**Acknowledgments:** The authors thank Allan Knopf and Ephraim Govere for assistance with the use of laboratory facilities.

**Conflicts of Interest:** The authors declare no conflict of interest. The funders had no role in the design of the study; in the collection, analyses, or interpretation of data; in the writing of the manuscript, or in the decision to publish the results.

**Disclaimer:** This work is not a product of the U.S. Government or the U.S. Environmental Protection Agency, and the author did not do this work in any governmental capacity. The views expressed are those of the author only and do not necessarily represent those of the U.S. Government or the EPA.

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
