# Peer review of "Simultaneous Extraction of Four Antibiotic Compounds from Soil and Water Matrices"

_separations, doi:10.3390/separations9080200_

Round 1

Reviewer 1 Report

The publication is interesting and the research topic is timely. Surely this work will be of great interest to analysts working with antibiotics and environmental samples. I have only a few minor comments that need to be addressed by the authors before publishing the paper.

The introduction should be changed. This paper deals with the extraction of antibiotics from environmental samples, so the focus should be on a literature review of methods used to date to extract these selected antibiotics. Please revise introduction

Lines 339-355 and 425-437: you have there some strange results since you have greater recoveries for dirty extracts (dirty FA, dirty ACN/H2O, dirty MeOH/H2O- table 3). That’s surprising and actually strange. The recoveries should be similar or lower, why they become greater compared to standard solutions? You should explain it in detail, not just generally like you’ve done that now in lines 432-437.

Have authors considered applying SPE as an additional purification step during antibiotics extraction from soil (after ASE extraction)?

Author Response

Reviewer Comment: The introduction should be changed. This paper deals with the extraction of antibiotics from environmental samples, so the focus should be on a literature review of methods used to date to extract these selected antibiotics. Please revise introduction

Author Reply:  Revised the introduction to provide a review of extraction methods commonly used. 

Reviewer Comment/Question:  Lines 339-355 and 425-437: you have there some strange results since you have greater recoveries for dirty extracts (dirty FA, dirty ACN/H2O, dirty MeOH/H2O- table 3). That’s surprising and actually strange. The recoveries should be similar or lower, why they become greater compared to standard solutions? You should explain it in detail, not just generally like you’ve done that now in lines 432-437.

Author Response:  Added in an explanation as to why the clean extracts may have lower recoveries than the dirty extracts. 

Reviewer Question:  Have authors considered applying SPE as an additional purification step during antibiotics extraction from soil (after ASE extraction)?

Author Response:  We do apply SPE after antibiotic extraction soil (after the ASE extraction). Adding an additional figure (Figure 1) to clearly show the analytical steps for soil extraction and aqueous extraction of antibiotics. 

Find updated manuscript attached with edits. 

Reviewer 2 Report

The study evaluates different extraction methods to extract four antibiotics from water and soil samples. The extraction procedure in each type of sample was optimized and compared with suitable results. Then, the selected method was developed and the results arisen thoroughly discussed. However, the study has two major issues: on one hand, only four antibiotics are selected, and in my opinion, this is really limited. It would be much more interesting a study with a more complete list of antibiotics. On the other hand, all the results derived from the extraction of soil samples are expressed as extraction recoveries and the matrix effect values were not even mentioned. Matrix effect should be evaluated and considered when such complex samples as soils are extracted and the extracts injected in LC-MS/MS. The discussion of the effectiveness of the extraction procedure with soil samples is pointless if the matrix effect is not considered. Apart from this, other issues are listed below:

-        The acronyms of the compounds should be described and used the second time in the abstract.

-        In the introduction (lines 96-101), it is explained the uses of SMX and TMP, but not the ones for the other two antibiiotics. Please, add this information to be consistent.

-        Section 2.6 should detail the ESI and QqQ conditions as well as the selection ions. Moreover, if acquiring in MRM mode, Q3 is working under SIM mode and not SCAN mode.

-        Along the text (for instance, lines 15, 342, …) it should be revised that the analytes are determined and the samples are analysed, and not in the other way around.

-        Add y axis in the graphics of figure 1.

-        The list of references are old-fashioned. At least some should be from the 10 (better 5)year ago   

Author Response

Reviewer Comment:  Limited study by only looking at four antibiotics

Author Reply: First, main reason for only looking at four antibiotics was due to funding. And even though only four antibiotics were analyzed, this study went beyond simply looking at extraction efficiencies for water and soil. It compared different types of HLB cartridges that people may want to use. It compared batch equilibrium with ASE extraction, so those who don't have access to high end equipment like an ASE machine now the effectiveness of batch equilibrium. It also looked at other factors that may impact the extraction:  how the soil extraction solution may impact subsequent SPE recoveries to clean up the solution prior to LC/MS-MS, grinding soil versus not grinding soil, freeze drying soil versus not freeze drying soil. All of which will provide valuable information for anyone wanting to extract these compounds from soils. 

Reviewer Comment:  Did not discuss matrix effects. 

Author Reply:  Added in discussion of matrix effects. 

Reviewer Comment: The acronyms of the compounds should be described and used the second time in the abstract.

Author Reply: Added description of acronyms in Abstract. 

Reviewer comment:  In the introduction (lines 96-101), it is explained the uses of SMX and TMP, but not the ones for the other two antibiiotics. Please, add this information to be consistent.

Author Reply: Added uses of other two antibiotics

Reviewer Comment:  Section 2.6 should detail the ESI and QqQ conditions as well as the selection ions. Moreover, if acquiring in MRM mode, Q3 is working under SIM mode and not SCAN mode.

Author Reply: Updated Section 2.6. 

Reviewer Comment:  Along the text (for instance, lines 15, 342, …) it should be revised that the analytes are determined and the samples are analysed, and not in the other way around.

Author Reply: Fixed wording throughout document. 

Reviewer Comment: Add y axis in the graphics of figure 1.

Author Reply: Added y axis.

Reviewer Comment:  The list of references are old-fashioned. At least some should be from the 10 (better 5)year ago   

Author Reply: Updated reference list. 

Find updated manuscript attached with edits. 

Reviewer 3 Report

The manuscript entitled "Simultaneous extraction of four antibiotic compounds from soil and water matrices" concerns the analysis from water and soil samples of 4 popular in the USA antibiotics. The analysis was carried out using 2 types of cartridges filled with sorbents possessing different chemical nature and thus ensuring a different way of interaction between the analytes and the sorbent.

The authors properly planned the research taking into account a number of factors influencing the obtained results. The provided descriptions of the laboratory work are accurate and clear.

In my opinion, the article may be published as it is, the Authors should only pay attention to the illegibility of the chemical structure of lincomycin in Table 1.

The second element that requires comment is the recovery exceeding 100% (118 +/- 5%) in Table 3

Author Response

Reviewer Comment:  Authors should only pay attention to the illegibility of the chemical structure of lincomycin in Table 1.

Author Reply:  Fixed the chemical structure of lincomycin in Table 1. 

Reviewer Comment:  The second element that requires comment is the recovery exceeding 100% (118 +/- 5%) in Table 3

Author Reply:  Added explanation about signal enhancement that can occur when working with complex matrices. 

Find updated manuscript attached with edits. 

Round 2

Reviewer 2 Report

The authors has addressed some of the reviewer's suggestions, and by this, the manuscript has improved.